# Analysis of potential genetic biomarkers and molecular mechanism of smoking-related postmenopausal osteoporosis using weighted gene co-expression network analysis and machine learning

**Shaoshuo Li**[1], **Baixing Chen**[2], **Hao Chen**[3], **Zhen Hua**[3], **Yang Shao**[3], **Heng Yin**[3], **Jianwei Wang**[ID][3]*

**1** Nanjing University of Chinese Medicine, Nanjing, P.R. China, **2** Department of Development and Regeneration, KU Leuven, University of Leuven, Leuven, Belgium, **3** Department of Traumatology & Orthopedics, Wuxi Affiliated Hospital of Nanjing University of Chinese Medicine, Wuxi, P.R. China

* wxwangjianwei1963@126.com

**Data Availability Statement:** All relevant data are within the manuscript and its Supporting Information files.

## Abstract

### Objectives

Smoking is a significant independent risk factor for postmenopausal osteoporosis, leading to genome variations in postmenopausal smokers. This study investigates potential biomarkers and molecular mechanisms of smoking-related postmenopausal osteoporosis (SRPO).

### Materials and methods

The GSE13850 microarray dataset was downloaded from Gene Expression Omnibus (GEO). Gene modules associated with SRPO were identified using weighted gene co-expression network analysis (WGCNA), protein-protein interaction (PPI) analysis, and pathway and functional enrichment analyses. Feature genes were selected using two machine learning methods: support vector machine-recursive feature elimination (SVM-RFE) and random forest (RF). The diagnostic efficiency of the selected genes was assessed by gene expression analysis and receiver operating characteristic curve.

### Results

Eight highly conserved modules were detected in the WGCNA network, and the genes in the module that was strongly correlated with SRPO were used for constructing the PPI network. A total of 113 hub genes were identified in the core network using topological network analysis. Enrichment analysis results showed that hub genes were closely associated with the regulation of RNA transcription and translation, ATPase activity, and immune-related signaling. Six genes (*HNRNPC*, *PFDN2*, *PSMC5*, *RPS16*, *TCEB2*, and *UBE2V2*) were selected as genetic biomarkers for SRPO by integrating the feature selection of SVM-RFE and RF.

**Funding:** This study was supported by the National Natural Science Foundation of China (No. 81873320, 81973878). The funders had no role in study design, data collection and analysis, decision to publish, or preparation of the manuscript.

**Competing interests:** The authors have declared that no competing interests exist.

## Conclusion

The present study identified potential genetic biomarkers and provided a novel insight into the underlying molecular mechanism of SRPO.

## 1. Introduction

Osteoporosis is a systemic skeletal disorder. This disease is highly prevalent worldwide and is characterized by bone microstructure degeneration, reduction in bone mineral density (BMD), leading to increased bone fragility and decreased bone strength [1, 2]. It is reported that almost 50% of postmenopausal women develop osteoporosis [3]. Furthermore, a third of postmenopausal women have bone fractures due to osteoporosis [4]. The estimated cost of managing postmenopausal osteoporosis (PMOP) and related fractures in the United States in 2015 was over USD 15 billion [5], and PMOP has become a major public health problem worldwide [6].

Multiple factors are involved in PMOP by affecting the function of osteoblasts and osteoclasts and regulating bone mineral homeostasis [7]. Estrogen secretion is decreased during menopause, resulting in the decline of ovarian function, increasing the risk of bone metabolic diseases [8, 9]. Estrogens modulate immune activity and the response of immune cells (T cells, B cells, and monocytes) to estrogen and its receptors [10]. Circulating B lymphocytes are strongly implicated in the pathogenesis of PMOP by producing cytokines that regulate the activity of osteoblasts and osteoclasts. In addition, the downregulation of MAPK3 and ESR1 in B cells decreases osteogenesis and increases osteoclastogenesis, demonstrating the importance of B cells in the etiology of PMOP [11].

Poor lifestyle habits are significant contributors to rapid bone loss in postmenopausal women [12]. In this context, smoking is a significant independent risk factor for osteoporosis ($P = 0.000$, OR = 1.911) [13]. Female smokers are almost twice as likely to have osteoporosis than non-smoking women [14]. Smoking may lead to changes in the microarchitecture of trabecular bone and reduces the ability of the skeletal muscle to resist mechanical load and stress [15]. Moreover, smoking may induce harmful changes in the immune system and cause diseases via the dysregulation of impaired B cells. Smoking-related postmenopausal osteoporosis (SRPO) is an emerging area of research that assesses changes in gene expression levels in postmenopausal smokers.

With the rapid development of high-throughput microarray technologies, the identification of genomic variations and biological mechanisms has improved our understanding of disease pathogenesis and treatment [16, 17]. Weighted gene co-expression network analysis (WGCNA) is widely used to analyze gene expression microarray data, identify functional gene modules, and discover relationships between gene modules and disease traits [18–20]. WGCNA screens genes and divides them into modules, which in turn are correlated with specific clinical phenotypes through Pearson correlation analysis. Machine learning algorithms have shown great promise in investigating the underlying relationship of high-dimensional data through supervised or unsupervised methods [21, 22]. Moreover, machine learning is useful to analyze high-dimension transcriptomic data and identify feature genes with biological significance [23–25]. However, no studies have analyzed genome variations in SRPO.

In this study, we performed a comprehensive analysis of gene expression patterns of circulating B cells from 20 postmenopausal female smokers with low or high BMD using bioinformatics and machine learning algorithms, including WGCNA, support vector machine-

recursive feature elimination (SVM-RFE), random forest (RF), protein-protein interaction (PPI) and functional analyses, and receiver operating characteristic (ROC) curve analysis. Six potential diagnostic biomarkers of SRPO were identified.

## 2. Materials and methods

### 2.1. Microarray data collecting and data preprocessing

The study flowchart is shown in Fig 1. The gene microarray dataset GSE13850 based on the Affymetrix Human Genome U133A (GPL96) platform, probe annotation files, and CEL files were downloaded from the Gene Expression Omnibus database (GEO, http://www.ncbi.nlm.nih.gov/geo/). Quantile normalization, background correction, and probe summarization of raw data were performed using the robust multiarray average (RMA) algorithm [26]. If one gene matched more than one probe, the maximum value of the probe was selected and calculated. The GSE13850 dataset provided data on gene expression in circulating B cells of 20 postmenopausal female smokers (10 with high BMD and 10 with low BMD).

### 2.2. Construction of the WGCNA network

Phenotype-correlated gene modules associated with SRPO were identified by WGCNA. The top 5,000 genes with the highest expression levels were used to construct the WGCNA network using the WGCNA package in R [20]. First, Pearson's correlation matrices for all pairs of genes were calculated. The pairwise correlation coefficient between the pair of gene m and gene n with significance ($S_{mn}$) was defined as $S_{mn} = |cor(m,n)|$. These correlation matrices were transformed into a weighted adjacency matrix using the power function $a_{mn} = power(S_{mn}, \beta) = |S_{mn}|^{\beta}$ [26]. According to the average connectivity degree and standard of approximate scale-free topology network, an appropriate soft-thresholding power $\beta$ was selected, and the adjacency matrix was transformed into a topological overlap matrix (TOM). TOM-based hierarchical clustering of gene modules was performed using the dynamic tree cut algorithm [27]. Gene modules with similar expression profiles were represented by different branches with appropriate colors, and the minimum module size was set as 40.

### 2.3. Correlation between gene modules and SRPO

The WGCNA algorithm uses module eigengene (ME) to evaluate relationships between gene modules and clinical traits. ME was defined as the major component computed by a principal component analysis that recapitulates the manifestation of genes from a specific module into a characteristic expression profile [28]. The Pearson correlation between ME and clinical traits was calculated to identify the module that was highly correlated with SRPO. The significance of Pearson correlation was assessed using a *t*-test, and the module with a *P*-value of less than 0.05 was considered to be significantly correlated with SRPO. Furthermore, gene significance (GS) and module membership (MM) were calculated for intramodular analysis. MM was the correlation between ME and the gene expression profile. GS was defined as the log10 transformation of the *P*-value (lg*P*) between gene expression and the clinical trait (GS = lg*P*). Module significance (MS) was defined as the average GS of all genes in a module. The module with the highest absolute MS was considered to be significantly correlated with SRPO. The module with the highest correlation with a clinical trait (osteoporosis) was selected as a research object.

### 2.4. Construction of PPI networks

PPI networks were constructed to evaluate the relationship among genes in the selected modules using the Search Tool for the Retrieval of Interacting Genes version 11 (STRING V11,

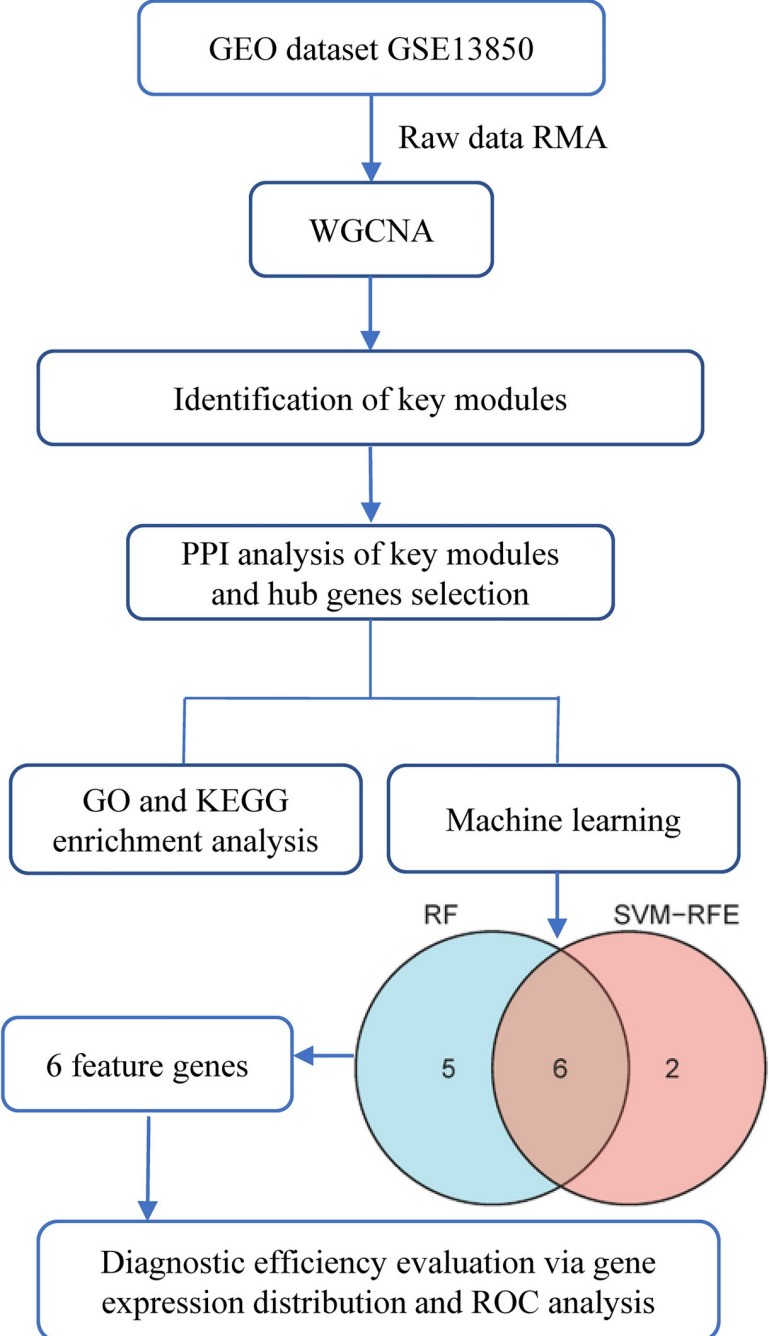

**Fig 1. Workflow of the present study.**

https://string-preview.org/). The confidence level was set as >0.4, and the network was visualized using Cytoscape version 3.8.2 [29]. Hub genes are highly interconnected nodes and may play important roles in the PPI network. A topological network analysis, including betweenness centrality (BC), closeness centrality (CC), and degree centrality (DC), for screening hub genes was performed using the CytoNCA plugin for Cytoscape [30].

## 2.5. Function and pathway enrichment analyses

Gene Ontology (GO) enrichment analysis and the Kyoto Encyclopedia of Genes and Genomes (KEGG) pathway enrichment analysis were performed using the clusterProfiler [31] package in R to describe the possible biological functions of hub genes. Three categories of biological process (BP), cellular component (CC), and molecular function (MF) were included in the GO terms. A Benjamini–Hochberg adjusted *P*-value of less than 0.05 was considered to indicate significantly enriched GO terms and KEGG pathways.

## 2.6. Machine learning for feature selection

Feature genes associated with SRPO were selected using SVM-RFE and RF. SVM-RFE was an efficient feature selection algorithm and had shown promising power in the analysis of the genomics [32], metabolomics [33], proteomics [34], etc. During the performance, SVM-RFE iteratively removed the features with the smallest weight from a rank until all features were excluded. In each iteration, the current SVM-RFE model was evaluated by k-fold cross-validation. After that, the classifier model with the highest accuracy was constructed, and the best variables were found [35]. The RF algorithm used the variables to construct numerous decision trees and generated the most accurate classes of variables to individual trees. RF has also been widely used for detecting disease biomarkers [36, 37]. The SVM-RFE model was built using the R package caret version 6.0–88. RF was applied using the randomForest package version 4.6–14. Ultimately, the common genes obtained using both SVM-RFE and RF were combined for further analysis.

## 2.7. Evaluation of the diagnostic efficiency

The ability of feature genes to differentiate between SRPO patients and non-osteoporosis postmenopausal smokers was evaluated by gene expression and ROC curve analyses. The predictive efficiency was measured in the control group (ten samples from postmenopausal smokers with high BMD) and the SRPO group (ten samples from postmenopausal smokers with low BMD). A Benjamini–Hochberg adjusted *P*-value of less than 0.05 were considered to indicate significant differences in gene expression. The ROC curve was created using the pROC package version 1.17.0.1 in R. The genes with an area under the ROC curve (AUC)>0.7 were considered to have good diagnostic performance.

# 3. Results

## 3.1. Data collection and WGCNA analysis

Gene expression data and clinical data from the GSE13850 dataset were downloaded from the GEO database. Following data processing, the top 5,000 genes in circulating B cells were collected, and the WGCNA network was constructed. Subsequently, an appropriate soft-thresholding power $\beta = 9$ was adopted due to the signed $R^2$ of the scale-free topology network was 0.85 (Fig 2).

Eight gene modules were obtained using the dynamic tree cut algorithm (Fig 3A and 3B). The correlation between each module and osteoporosis was assessed by calculating the module–trait relationship and MS. First, the Pearson correlation between the ME of each module and osteoporosis was calculated and shown in the module–trait relationship heatmap (Fig 3C and Table 1). The blue module (module–trait relationships = 0.88, *P*-value = 7e-07) had the highest association with osteoporosis. After that, the MS of each module was calculated. We found that the blue module had the highest MS among all selected modules (Fig 3D). Hence, the 1078 genes in the blue module were significantly associated with SRPO, and these genes

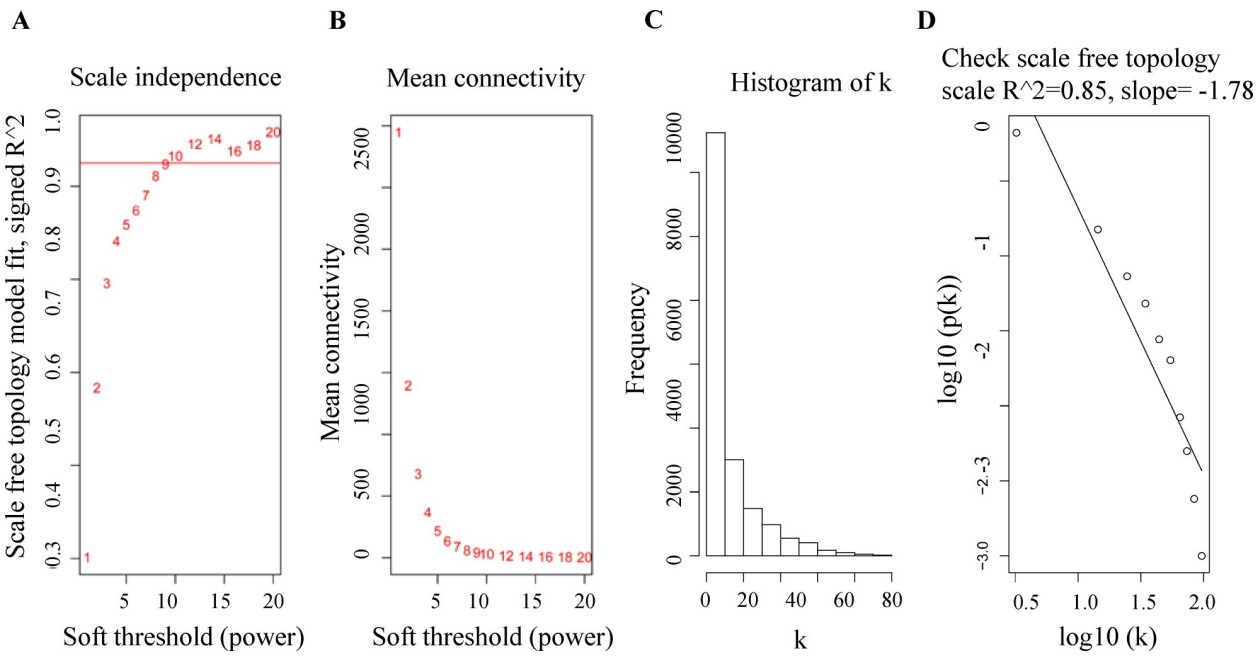

**Fig 2. Construction of the weighted gene co-expression network of gene modules.** (A) Analysis of the scale independence for the appropriate soft-thresholding power β. (B) Analysis of the mean connectivity for the appropriate soft-thresholding power β. (C) Histogram of connectivity distribution with an appropriate β = 9. (D) Checking the scale-free topology with an appropriate β = 9.

were selected for subsequent analysis in the PPI network. The clustering heatmap of the ME of the blue module and the scatterplots of GS vs. MM are presented in Fig 3E and 3F.

## 3.2. Construction of the PPI network and enrichment analysis of hub genes

After removing the disconnected nodes, there were 998 nodes and 10940 edges in the constructed PPI network for genes in the blue module (Fig 4A). According to topological network analysis, PPI nodes are considered significant targets if the DC is greater than two-fold the median DC [38]. Thus, DC > 28 was set as the threshold, and significant nodes were identified to generate a subnetwork. Then, nodes where BC and CC values were greater than the median in the subnetwork (BC>158.81, CC>0.48) were considered a new core network containing hub genes. The core network containing 113 hub genes (nodes) and 1831 edges is shown in Fig 4B.

Functional enrichment analysis was performed to improve biological understanding of the hub genes identified in the PPI network. Regarding biological processes, GO analysis showed that hub genes were mainly involved in the regulation of mRNA transcription, regulation of cell cycle, protein targeting, and cellular response to hypoxia (Fig 5A). In the cellular component analysis, hub genes were mainly associated with ribosomal subunits, methylosome, and proteasome complexes (Fig 5B). Significantly enriched molecular functions were translation regulation, ATPase activity, hormone receptor binding, and protein binding (Fig 5C). KEGG pathway enrichment analysis showed that ribosome, apoptosis, mitophagy, HIF-1 signaling pathway, NF-kappa B signaling pathway, Th17 cell differentiation, and B cell receptor signaling pathway were the most significant processes in SRPO (Fig 5D).

## 3.3. Identification of feature genes using machine learning algorithms

Machine learning classification algorithms are being increasingly used to predict feature genes associated with diseases from the noise background. SVM-RFE and RF were used to predict

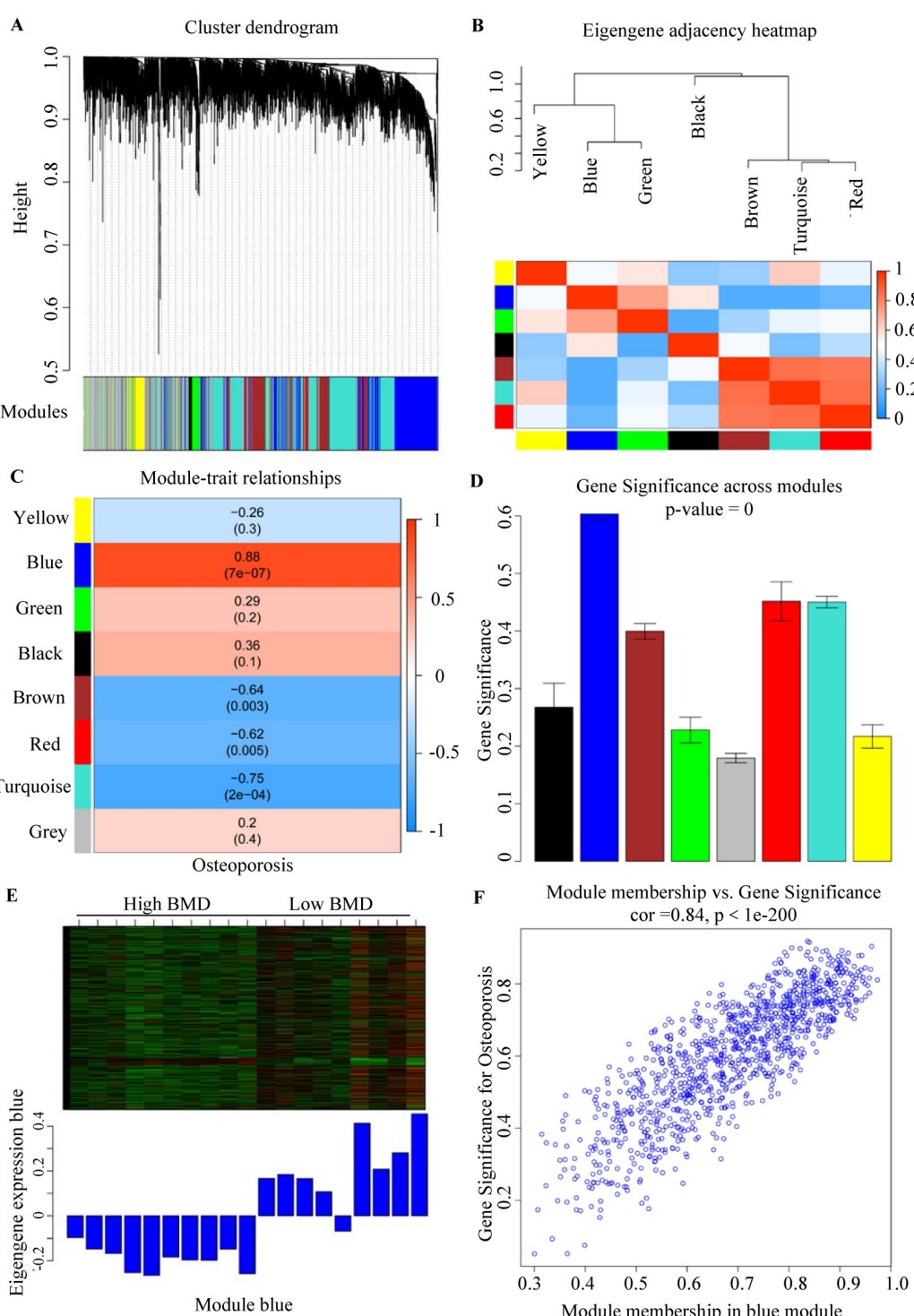

**Fig 3. Identification of significant gene modules correlated with osteoporosis.** (A) Cluster dendrogram of representative gene modules. (B) Clustering heatmap of module eigengenes. (C) Relationships of module eigengenes and osteoporosis. The number in the square at the top of each row is the correlation coefficient, and *P*-values are shown below. (D) Gene significance across modules. (E) Heatmap and bar graph of the eigengenes in module blue. (F) Scatterplot of gene significance vs. module membership in the blue module.

**Table 1. Correlation between modules and smoking-related postmenopausal osteoporosis.**

| Modules | Gene count | Correlation | P-value |
| --- | --- | --- | --- |
| Blue | 1078 | 0.88 | 7e-07 |
| Turquoise | 1728 | -0.75 | 2e-04 |
| Brown | 771 | -0.64 | 0.003 |
| Red | 78 | -0.62 | 0.005 |
| Black | 60 | 0.36 | 0.1 |
| Green | 164 | 0.29 | 0.2 |
| Yellow | 225 | -0.26 | 0.3 |
| Grey | 896 | 0.2 | 0.4 |

feature genes associated with SRPO. First, an SVM-RFE classifier (Core: svmliner; Cross: 10-fold cross-validation; soft-margin; tuning parameter C = 1) was established based on 113 hub genes. Data from the control and SRPO groups were randomly divided into ten equal portions (training set: 9; test set: 1). During each of the ten iterations, SVM-RFE was applied to the training set to train the classifier with the selected features, and the trained classifier was applied to the test set to assess prediction accuracy. Then, the predictions from the ten iterations were combined to evaluate the accuracy of the classifier. Eight feature genes were validated using SVM-RFE (Fig 6A). Similarly, feature genes were screened by 10-fold cross-validation using RF algorithm. The RF classifier showed a least out-of-bag (OOB) error with the top 11 feature genes (Fig 6B). After integrating feature genes from SVM-RFE and RF, six feature genes closely associated with SRPO were obtained: *HNRNPC, PFDN2, PSMC5, RPS16, TCEB2*, and *UBE2V2* (Fig 6C).

## 3.4. Diagnostic efficiency of feature genes

The difference in expression pattern of the six feature genes between the SRPO and control groups was assessed. Gene expression was downregulated in the SRPO group, except for *UBE2V2* (Fig 7A). To identify if the feature genes influence SRPO diagnosis independently, ROC analysis was performed. The results showed that the ability of these genes to diagnose SRPO was high, with an AUC>0.9 (Fig 7B).

As an RNA-binding protein, heterogeneous nuclear ribonucleoprotein C (*HNRNPC*) is well known for regulating mRNA metabolism and RNA expression, splicing, and translation [39, 40]. In addition, *HNRNPC* regulates N6-methyladenosine (m6A) RNA methylation, which is crucial to neurogenesis, embryonic development, stress responses, and tumorigenesis [41, 42]. *TCEB2* (also known as *ELOB*) encodes the protein elongin B, a subunit of the transcription factor B complex and an adapter protein in the proteasomal degradation of target proteins through E3 ubiquitin ligases [43]. Proteasome 26S subunit ATPase 5 (*PSMC5*) interacts with several transcription factors, including nuclear hormone receptors, p53, c-Fos, and the basal transcription complex [44]. Moreover, *PSMC5* plays a proteasome-independent role in DNA repair, chromatin remodeling, and transcription activation and elongation [45, 46]. *PFDN2* is a component of β subunits of the URI prefoldin-like complex, which plays a critical role in maintaining cellular homeostasis [47]. Ubiquitin-conjugating enzyme E2 variant 2 (*UBE2V2*) mediates the transcriptional activation of target genes and controls cell differentiation, cell cycle, and DNA damage response [48]. Ribosomal protein S16 (*RPS16*), the basic component of the 40S ribosome, was reported to be associated with the defective mitochondrial translation [49]. These feature genes were closely associated with RNA transcription and translation, and important cellular activity in SRPO.

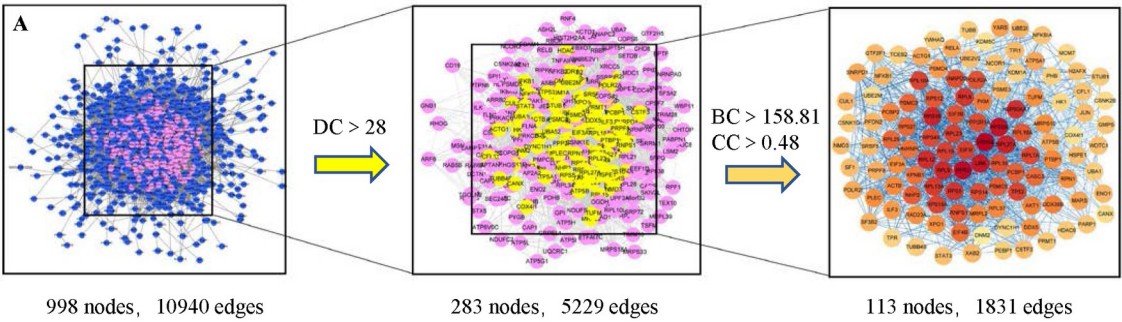

998 nodes, 10940 edges 283 nodes, 5229 edges 113 nodes, 1831 edges

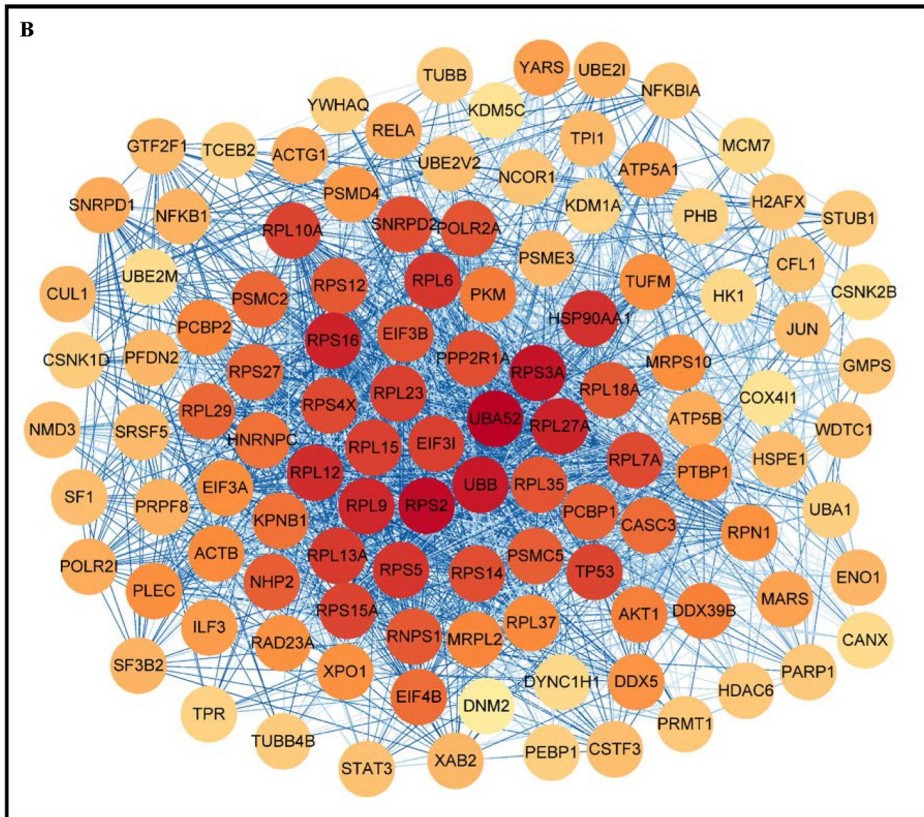

**Fig 4. Protein-Protein Interaction (PPI) network of genes from the blue module.** (A) Screening of hub genes. The screening criteria were degree centrality>28, betweenness centrality>158.81, and closeness centrality>0.48. (B) Core PPI network with 113 hub genes and 1831edges. The color of the nodes represented the value of degree. The darker (red) the color, the higher the degree.

## 4. Discussion

There is increased public awareness of the harmful effects of exposure to cigarette smoking. However, although substantial progress has been made in tobacco control, cigarette smoking remains one of the most challenging global health issues to date [50, 51]. Postmenopausal smokers are at an increased risk of developing osteoporosis and osteoporotic fractures than non-smoking females [52]. Moreover, smoking-induced genetic alterations influence hormone secretion and bone metabolism in women [53, 54]. The molecular mechanism of occurrence and development of SRPO is incompletely understood, and identifying new biomarkers for SRPO diagnosis and treatment is crucial.

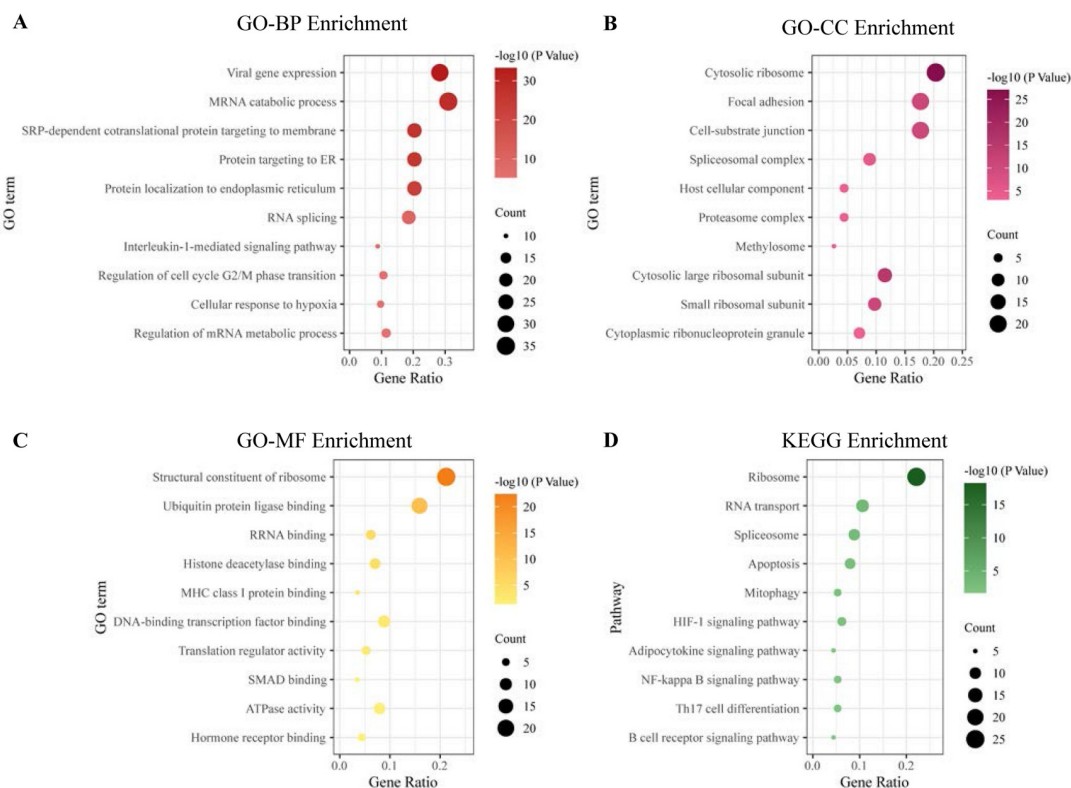

**Fig 5. Functional enrichment analysis of hub genes.** (A-C) Gene ontology enrichment analysis. (D) Kyoto Encyclopedia of Genes and Genomes pathway enrichment analysis. BP, biological process; CC, cellular component, MF, molecular function.

We determined the gene expression profiles in circulating B cells from 20 postmenopausal smokers with low or high BMD. First, WGCNA was performed to select the gene modules with the strongest correlation with SRPO. Then, 1078 genes in the selected module were used to construct a PPI network. Topological network analysis identified a core PPI network and 113 hub genes. Functional enrichment analysis showed that these hub genes were closely associated with the development of SRPO via the control of several biological processes, including the regulation of RNA transcription and translation, hormone receptor binding, and NF-kappa B signaling pathway. Previous studies have shown that these biological processes and signaling pathways are implicated in bone metabolism and osteoporosis [55, 56]. The risk of missing important features was minimized by incorporating genes using two machine learning algorithms. SVM-RFE and RF were performed to screen six characteristic variables from these hub genes. Diagnostic efficiency analysis showed that the genes *HNRNPC*, *PFDN2*, *PSMC5*, *RPS16*, *TCEB2*, and *UBE2V2* were potential biomarkers for SRPO.

In a cigarette smoke-induced chronic obstructive pulmonary disease (COPD) animal model, *HNRNPC* was overexpressed in the lungs of cigarette smoke-exposed mice [57]. The dysregulation of *HNRNPC* is associated with telomere shortening in lung cells and circulating lymphocytes, impairing lung function and increasing COPD severity and mortality [58, 59]. In addition, the dysregulation of *HNRNPC* may increase the expression of the urokinase plasminogen activator receptor, resulting in inflammation and immune activation [60]. *TCEB2* plays an essential role in the development of acquired resistance to anti-angiogenic therapy in ovarian cancer cells via suppressing VEGF-A expression and promoting HIF-1α degradation [61]. The vascularization of bone tissue is tightly linked with bone formation in a spatial and

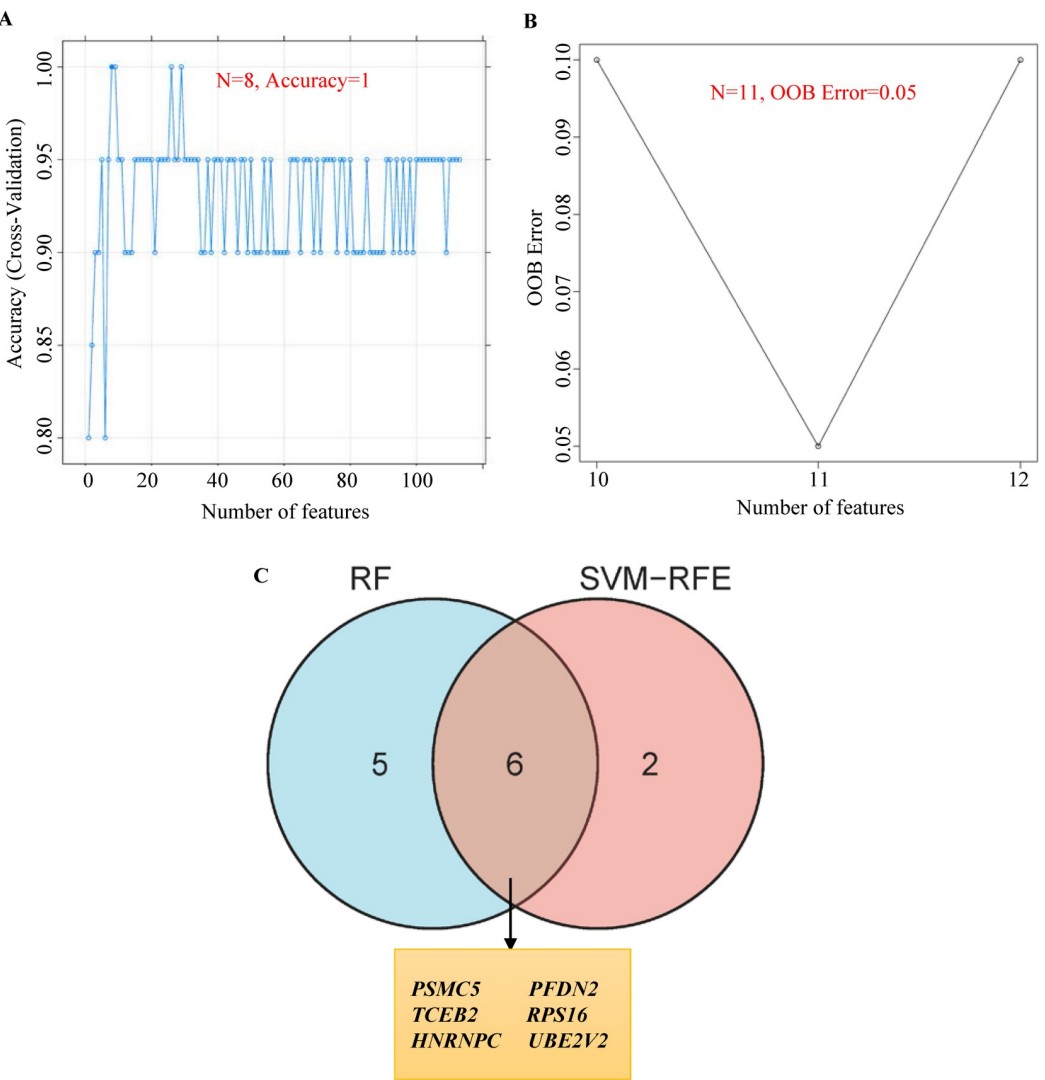

**Fig 6. Feature genes selection.** Using support vector machine-recursive feature elimination (SVM-RFE) (A) and random forest (RF) (B). (C) Venn plot of feature genes selected by RF and SVM-RFE.

temporal relationship known as angiogenesis-osteogenesis coupling [62]. Many factors, including HIF-1 and VEGF, regulate bone vascularization and angiogenic-osteogenic coupling in the bone microenvironment [63]. In this respect, the dysregulation of *TCEB2* may contribute to SRPO by impairing this coupling. *PSMC5* regulates ERK1/2 signaling transmission by remodeling the Shoc2 scaffold complex [64]. The activation of the ERK1/2 signaling cascade regulates the function of osteoblasts and osteoclasts, promoting inflammation and osteogenesis [65, 66]. *PFDN2* is closely associated with several diseases, such as Alzheimer's disease, colon cancer, and myelodysplastic syndromes, via different mechanisms [67–69]. The presence of antibodies against *PFDN2* is associated with an increased risk of type 2 diabetes through autoimmune activation and/or pro-inflammatory signals, which are involved in the regulation of bone homeostasis [70]. *UBE2V2* contributes to the development and progression of many cancers, including prostate, oropharyngeal, and breast cancers, via promoting cell proliferation, suppressing cell apoptosis, and regulating immune signaling [71–73]. Moreover, *UBE2V2* is an independent prognostic indicator for lung adenocarcinoma, which is closely

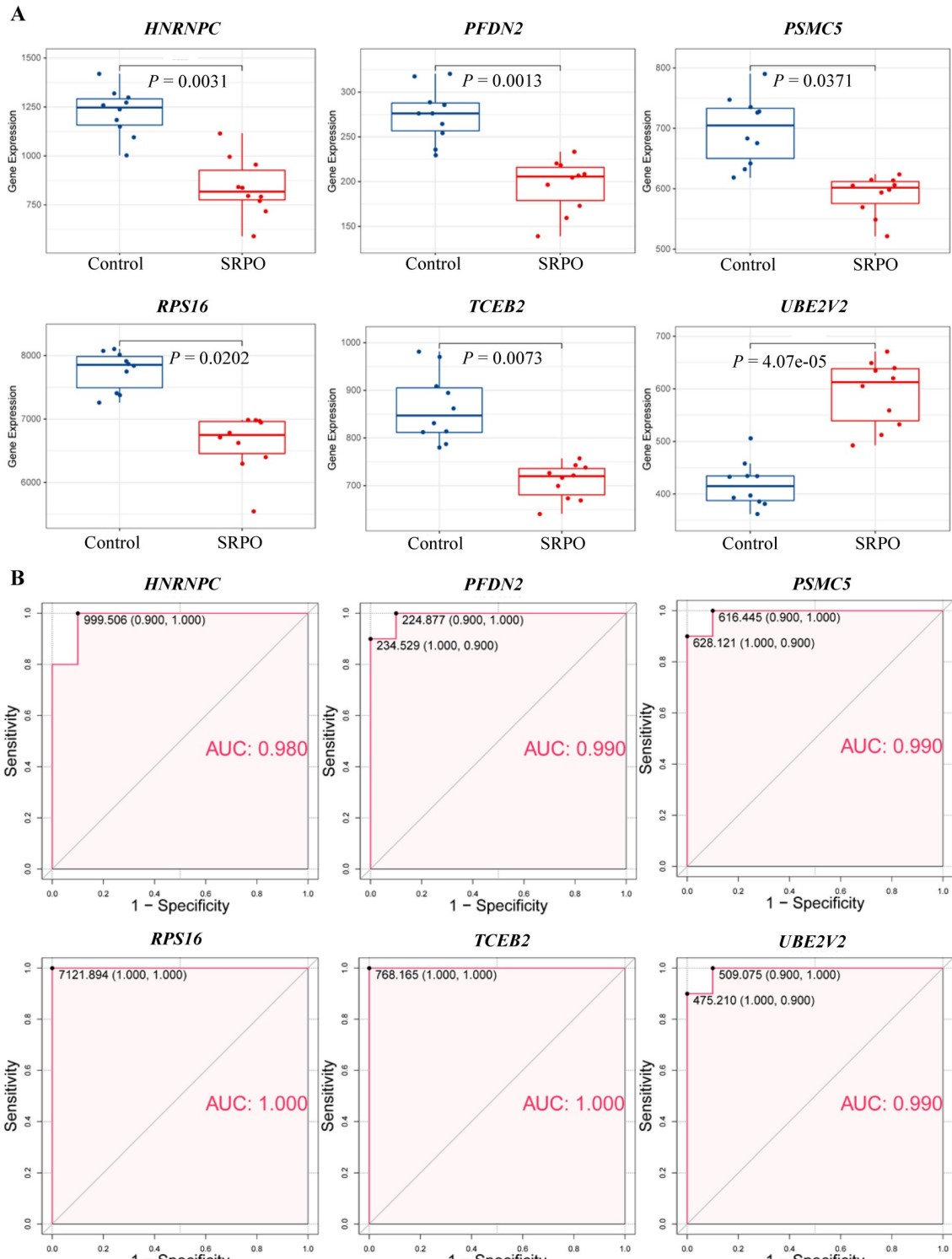

**Fig 7. Diagnostic efficiency evaluation of feature genes.** (A) Gene expression of six feature genes (*HNRNPC*, *PFDN2*, *PSMC5*, *RPS16*, *TCEB2*, and *UBE2V2*) in women with smoking-related postmenopausal osteoporosis and controls. (B) Receiver operating characteristic curve analysis.

related to the mutational processes of cigarette smoking [74, 75]. *RPS16* contributes to facilitate tumor progression of glioma via the PI3K/AKT signaling [76]. Previous studies have indicated that the PI3K/AKT signaling pathway is an important factor in the occurrence of osteoporosis by regulating the activity of osteoblasts and osteoclasts [77, 78].

WGCNA can identify genes with clinical significance and cluster genes associate with pathological processes based on medical and biological background. Machine learning algorithms have shown objective assessment and optimal accuracy in feature selection. The present study is the first to perform a comprehensive strategy of machine learning algorithms and WGCNA to identify potential biomarkers of SRPO. Although our results are consistent with the literature, the reliability of this study needs to be verified by further experiments. This study has limitations. First, the smoking history, frequency, and status of individuals in the study were not well known, which might cause uncontrolled factors in data analysis. Second, the identified biomarkers were not functionally and externally validated. Third, the small sample size may have limited the power of the study. Additional studies on the association of these biomarkers with SRPO are warranted.

## 5. Conclusion

The present study identified six genes (*HNRNPC*, *PFDN2*, *PSMC5*, *RPS16*, *TCEB2*, and *UBE2V2*) as potential biomarkers for SRPO using WGCNA and machine learning algorithms, providing a novel insight into the diagnosis and treatment of SRPO. However, these biomarkers need to be validated by clinical trials.

## Supporting information

**S1 Data. Gene expression of samples.**
(XLSX)

**S2 Data. Genes in each module.**
(XLSX)

**S3 Data. PPI network.**
(XLSX)

**S4 Data. Results of CytoNCA analysis.**
(XLSX)

**S5 Data. GO enrichment of hub genes.**
(XLSX)

**S6 Data. KEGG pathway enrichment.**
(XLSX)

**S7 Data. Feature selection of machine learning.**
(XLSX)

## Acknowledgments

We would like to thank TopEdit (www.topeditsci.com) for its linguistic assistance during the preparation of this manuscript.

## Author Contributions

**Conceptualization:** Shaoshuo Li.

**Data curation:** Shaoshuo Li, Baixing Chen.

**Formal analysis:** Shaoshuo Li, Baixing Chen, Hao Chen.

**Investigation:** Zhen Hua, Yang Shao.

**Methodology:** Shaoshuo Li.

**Project administration:** Heng Yin, Jianwei Wang.

**Software:** Shaoshuo Li.

**Supervision:** Shaoshuo Li.

**Validation:** Shaoshuo Li.

**Visualization:** Shaoshuo Li.

**Writing – original draft:** Shaoshuo Li, Baixing Chen.

**Writing – review & editing:** Shaoshuo Li, Heng Yin, Jianwei Wang.

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
