## [Decision Letter · Decision Letter 0]

26 May 2021

PONE-D-21-08513

Investigation of Potential Genetic Biomarkers and Molecular Mechanism of Smoking-related Postmenopausal Osteoporosis by Using WGCNA and Machine Learning

PLOS ONE

Dear Dr. Wang,

Thank you for submitting your manuscript to PLOS ONE. After careful consideration, we feel that it has merit but does not fully meet PLOS ONE’s publication criteria as it currently stands. Therefore, we invite you to submit a revised version of the manuscript that addresses the points raised during the review process.

Both reviewers raised substantial technical concerns. All of these need to be suitably addressed in a revised manuscript.

We look forward to receiving your revised manuscript.

Kind regards,

Jishnu Das, Ph.D.

Academic Editor

PLOS ONE

Journal Requirements:

2. Please include your tables as part of your main manuscript and remove the individual files. Please note that supplementary tables should be uploaded as separate "supporting information" files.

Reviewers' comments:

Reviewer's Responses to Questions

**Comments to the Author**

1. Is the manuscript technically sound, and do the data support the conclusions?

Reviewer #1: No

Reviewer #2: Partly

2. Has the statistical analysis been performed appropriately and rigorously? 

Reviewer #1: No

Reviewer #2: No

3. Have the authors made all data underlying the findings in their manuscript fully available?

Reviewer #1: Yes

Reviewer #2: No

4. Is the manuscript presented in an intelligible fashion and written in standard English?

Reviewer #1: No

Reviewer #2: No

5. Review Comments to the Author

Reviewer #1: Review of "Investigation of Potential Genetic Biomarkers and Molecular Mechanism of Smoking-related Postmenopausal Osteoporosis by Using WGCNA and Machine Learning" by J. Wang.

The author has presented a bioniformatic pipeline to identify genetic biomarkers for smoking related postmenopausal osteoporosis. The are a few questions I have that I would like the authors to address.

1. The author seems to have mixed analytical methods with subjective heuristics, which is concerning. For example, based on correlation analysis the author identifies the yellow module as showing a significant correlation between both smoking and osteoporosis. However, this identification seems subjective as the black module also seems to be significant. Futhermore, since this data is not RNAseq data but microarray data, which generaly profiles targeted transcripts, it is quite possible that the black module will also identify some relevant biomarkers. The author, therefore, needs to carefully, methodically and quantitatively justify the selection of the relevant module.

2. Following the above point the author should clarify how the correlation between smoking/osteoporosis phenotype and eigengene of the different modules is computed and how the significance is computed. Additionally, in the Results section the author mentions correlation between eigenvalue (instead of eigengene) and the phenotype. The authors should clarify this confusion.

3. The author uses a recursive feature elimination approach. In the context of statistical regression, such stepwise approaches have been shown to be suboptimal. Why does the author think that this approach is a good option, especially given that many of the microarray genes might already be pre-selected and correlated. Clarification would be quite helpful.

4. What type of SVM did the author use: linear/non-linear, soft/hard margin, nu/C parameterization and why?

5. The author should clarify their validation approach: what was the training set, what was the testing set, and how did the control set relate to them.

6. It remains unclear to me as to why the author has generated ROCs and related AUC only for individual gene expressions. The performance is quite poor, and if one looks at the shape of the ROC, the sensitivity and specificity are not good either. It might be better to generate ROC curves for gene expression combinations.

7. In the abstract the author has mentioned the "yellow module" as the clinically significant module. This seems to be a highly unusual way to present ones results. Nobody reading the abstract will understand what a yellow module is. The author needs to provide a better description.

8. The writing needs to be improved. For example, 'According to recently scientific discoveries, ...', 'With the rapid development of high-throughput microarray technologies, identification of meaningful genomic variations and investigating biological mechanisms have contributed great effort ....', etc.

Reviewer #2: The overall analytical pipeline (WGCNA+ML) seems promising to help identify potential genetic biomarkers of SRPO, but the validity and robustness of the results presented in this paper need further examination.

Major

1. Identification of significant modules and analysis of module-trait relationship – It was not clear whether proper multiple-testing correction has been applied. This appeared to be critical as the significance of the two modules nominated in the Module-Osteoporosis analysis is marginal (Yellow p=0.03, Green p=0.02). And the significance of the Yellow in Module-Smoking relationship will also become marginal if Bonferroni is applied to correct 11 tests.

2. Construction of PPI network – The network and sub-network built in this paper seemed to be very dense. This was likely resulted from the authors including all sorts of protein linkages. Restricting to only “experimentally determined interactions” may better clarify the biological meaning of the network and the subsequent results. In particular, the inclusion of “gene neighborhood interactions”, “text-mining interactions”, and “gene co-occurrence interactions” lacks biological context.

3. Function and pathway enrichment analysis – 1) Statistical significance again was not properly defined here. While gene set enrichment tools usually provide adjusted p-values (or q-values), the authors stated that “The p-value < 0.05 was considered to indicate a significant difference for GO terms and KEGG pathways”. 2) Statistical significance (Adjusted P-value or Odds) would be preferred in the figure, instead of “gene number”. 3) It is of little use to list all the enriched terms in the main text, instead, the authors should investigate further and explain the biological meaning of these terms and more importantly, how they are relevant to the phenotype SRPO. This is essential and is part of be the meat of this paper, as the authors claimed that this work could provide novel insights into the molecular mechanism of SRPO.

4. Identification of feature genes using machine learning method – It was not sufficiently justified why the specific two ML methods were chosen. Also, how the recursive feature elimination analysis was performed and how it can benefit the model was not clearly explained. Plus, while prioritizing the genes by their overlaps seems justifiable, more detailed comparison of the two sets of genes is desired to give a more comprehensive view of the results (e.g., Are those non-overlapping genes also functionally relevant? Does one gene set appear to be more relevant than the other? Why RF gave many more genes than SVM – is it more powerful or less accurate? Would it be reasonable to use the union instead of intersection to implicate more genes?).

5. Data validation – 1) The selection of control group: “10 samples of postmenopausal non-smoking females with high BMD” was used, while alternatively, the other group “10 samples of postmenopausal non-smoking females with low BMD” can be used to better dissect the relationship between the implicated genes and “smoking-related” postmenopausal osteoporosis (rather than the general postmenopausal osteoporosis). 2) Multiple-testing correction is expected for the gene expression analysis; if Bonferroni, genes ATP5G1 and RPL26L1 will not surpass the threshold. 3) Description and discussion about the functional significance of the implicated genes are expected to follow the results here. The authors had lengthy paragraphs on this in the Discussion section, which should be moved up, expanded in depth, while rephrased more concisely.

Minor

1. Accurate references need to be cited when linking the implicated genes/pathways to SRPO if the relevant conclusions are not drawn by this paper (e.g., “It has been suggested that aging and increasing in reactive oxygen species (ROS) may be the proximal culprits for osteoporosis”, “ROS can influence the generation and survival of osteoclasts and osteoblasts” …).

2. Grammar needs to be revised (small errors like "Osteoporosis is one of the most common systemic skeletal disorder" occurs occasionally).

3. Language needs to be polished into a more scientific fashion (for example, in “… performed a heatmap and bar graph of ...” and “…scatterplots of GS vs. MM of module yellow of the two phenotypes were performed …”, specific analyses/statistical tests should be described rather than the types of graphs).

4. Data availability – key results should be complied as supplementary files for others to use (e.g., module information, PPI network, pathways, and relevant statistical results, etc.).

6. PLOS authors have the option to publish the peer review history of their article (what does this mean?). If published, this will include your full peer review and any attached files.

Reviewer #1: No

Reviewer #2: No

---

## [Author Response · Author response to Decision Letter 0]

2 Aug 2021

Thank you for your letter and the reviewers’ comments on our manuscript entitled " Investigation of Potential Genetic Biomarkers and Molecular Mechanism of Smoking-related Postmenopausal Osteoporosis by Using WGCNA and Machine Learning" (ID：PONE-D-21-08513). Those comments are very helpful for revising and improving our paper, as well as the important guiding significance to other researches. We have studied the comments carefully and made corrections which we hope meet with approval. The main corrections are in the manuscript and the responses to the reviewers’ comments are as follows (the replies are highlighted in blue). In addition, we have added several important supporting information.

Replies to the reviewers’ comments: 

Reviewer #1:

1.Response: According to your suggestion, we rewrote the “Correlation between gene modules and SRPO” in the “Methods and materials” and extended the reason why we chose blue module as the research object. In this study, we aimed to investigate potential genetic biomarkers and the underlying molecular mechanism of smoking-related postmenopausal osteoporosis. We collected the gene expression data of 20 postmenopausal female smokers (10 with high BMD and 10 with low BMD). Smoking was a common exposure for these people. The difference between these people was whether osteoporosis occurs or not, which may be related to the smoking‐induced genetic alterations. The genetic biomarkers of SRPO should differentiate SRPO patients and non-osteoporosis postmenopausal female smokers. So, we used WGCNA to identify the gene modules that highly correlated to the clinic trait (osteoporosis). After the construction of a weighted gene co-expression module network, module gene (ME) was used to evaluate the possible relationship of gene module with clinic trait. The Pearson correlation between ME and clinic trait was calculated to identify the module that was highly correlated to osteoporosis. The t-test was used to measure the significance of the Pearson correlation, and the module with a P-value less than 0.05 was considered to have a significant correlation to SRPO. Besides, the module significance (MS) of each module was calculated. The module with the highest absolute MS value was indicated to be significantly correlated to SRPO. As a result, the blue module (module–trait relationships = 0.88, P-value = 7e-07) was found to have the highest association with osteoporosis. And the blue module was found to have the highest MS value among all of the selected modules. Based on these results, we chose the blue module as the research object. 

2.Response: Thanks for the suggestion of the reviewer, we described the calculation method in detail in the “Methods and materials”. The WGCNA algorithm used module eigengene (ME) to evaluate the possible relationship of gene module with clinical trait. ME was defined as the major component computed by means of principal component analysis that recapitulates the manifestation of genes in a specific module into a single characteristic expression profile. The Pearson correlation between ME and clinical trait (osteoporosis) was calculated to identify the module that was highly correlated to SRPO. The t-test was used to measure the significance of the Pearson correlation, and the module with a P-value less than 0.05 was considered to have a significant correlation to SRPO. We are sorry to make a confusion of eigengene and eigenvalue. We have made corrections according to the Reviewer’s comments. In the module–trait relationships heatmap, the correlation between module eigengene and clinic trait was computed.

3.Response: According to the suggestion of the reviewer, we rewrote the “Machine learning for feature selection” in the “Methods and materials” and extended the reason why we used the recursive feature elimination approach for the feature genes selection. In this study, we used support vector machine-recursive feature elimination (SVM-RFE), and random forest (RF) algorithm to select feature genes from hub genes that correlated to SRPO. The recursive feature elimination approach removed one feature with the smallest weight iteratively to a feature rank until all the features had been removed. In each iteration, the current SVM-RFE model was evaluated by the k-fold cross-validation. Finally, the classifier model with highest accuracy was constructed and the best variables were found. The support vector machine combined with recursive feature elimination algorithm had shown promising power in the analysis of the genomics, metabolomics, and proteomics. For instance, in the previous study, SVM-RFE had helped identifying biomarkers correlated with breast cancer prognosis (Li J, et.al Tumor Characterization in Breast Cancer Identifies Immune-Relevant Gene Signatures Associated with Prognosis. Front. Genet. 10:1119.). Besides, the main advantage of microarray chips is that they can simultaneously examine the expression of thousands of genes on a large scale and comprehensively. For example, through gene microarrays, genes that may be affected by a disease can be found in a short period of time and serve as biomarkers for early diagnosis. Therefore, we believed that the SVM-RFE machine learning algorithm is a good option for feature genes selection and we can use SVM-RFE algorithm to identify reliable biomarkers for SRPO. 

4.Response: We are very sorry for our neglect to describe the process of identification of feature genes using machine learning algorithms in detail. We have rewritten this part in the results according to the Reviewer’s suggestion. Firstly, an SVM-RFE classifier (Core: svmliner; Cross: 10-fold cross-validation; soft-margin; tuning parameter C = 1) was established based on the 113 hub genes. The data of control group and SRPO group was randomly divided into 10 equal portions (training set: 9, test set: 1). During each of the ten iterations, the SVM-RFE algorithm was performed on the training set to train the classifier with the selected features. Next, the trained classifier was applied to the test set for prediction. Then, the predictions from all 10 iterations were combined to evaluate the accuracy of the classifier. By the SVM-RFE algorithm, we validated a set of 8 feature genes (Fig. 6A). In our SVM-RFE model, the tuning parameter C (also known as Cost) was set as default (C=1). And we used the soft-margin in our SVM-RFE for the reason that soft-margin SVM could choose decision boundary that has non-zero training error even if dataset is linearly separable, and is less likely to overfit. Besides, during the feature selection, we tried several types of SVM models，including linear, radial, sigmoid. Finally, the SVM-RFE classifier (Core: svmliner; Cross: 10-fold cross-validation) with these 8 feature genes revealed the highest predict accuracy, and this classifier was selected by the SVM-RFE algorithm. The prediction result based on the selected SVM-RFE classifier has been provided as a supplementary file. 

5.Response: The comments are very helpful for revising and improving our paper, and we have made correction according to the Reviewer’s comments. We rewrote the “Data validation” and described this section as “Evaluation of the diagnostic efficiency”. After integrating the results of feature selection from SVM-RFE and RF, we obtained six feature genes highly correlated to SRPO. The ability of feature genes to differentiate between SRPO patients and non-osteoporosis postmenopausal smokers was evaluated by gene expression and ROC curve analyses. The predictive efficiency was measured in the control group (ten samples from postmenopausal smokers with high BMD) and the SRPO group (ten samples from postmenopausal smokers with low BMD). As smoking was the common exposure for these people, the occurrence of SRPO between them may be highly related to the smoking‐induced genetic alterations. The biomarkers of SRPO should distinguish SRPO patients and non-osteoporosis postmenopausal female smokers precisely. As a result, the expression pattern of selected feature genes was significantly differentiated between the two groups. Moreover, these feature genes reached great diagnosis value with an AUC＞0.9, respectively. Furthermore, we committed certain limitations of this study in the discussion. There was little research on genomic variation in SRPO for now. To our best knowledge, this study is the first to investigate potential genetic biomarkers of SRPO. However, functional validation and external validation of these biomarkers were lacked in this study. The reliability of this study still needs to be verified by further experiments. 

6.Response: We are very sorry for our neglect to explain the purpose of the ROC analysis for individual gene expressions of the feature genes. We rewrote the “Data validation” and described this section as “Evaluation of the diagnostic efficiency” and we extended the reason why we performed ROC analysis for the feature genes in this section. In order to identify if the feature genes influence the SRPO diagnosis independently, we performed receiver operating characteristic (ROC) curve analysis for the selected feature genes on individual gene expression. As a result, the six feature genes reached great diagnosis value with an AUC＞0.9, respectively. Moreover, the shape of the ROCs indicated great sensitivity and specificity of these feature genes in SRPO diagnosis. It might be a good idea to generate ROC analysis for gene expression combination of all the feature genes. But the above ROC results of individual feature gene expressions have proven that the feature genes can influence the SRPO diagnosis independently. 

7.Response: We are sorry that our English writing level has troubled the reviewers. We have submitted our manuscript to TOPEDIT (https://www.topeditsci.com/) for polishing as suggested by the reviewers. Correction have been made as follow. Eight highly conserved modules were detected in the WGCNA network, and the genes in the module that was strongly correlated with SRPO were used for constructing the PPI network.

8.Response: We apologize for the language problems in the original manuscript. We have submitted our manuscript to TOPEDIT (https://www.topeditsci.com/) for polishing as suggested by the reviewers. 

Reviewer #2:

Major

1.Response: According to your suggestion, we rewrote the “Correlation between gene modules and SRPO” in the “Methods and materials”. In the revised paper, we aimed to investigate potential genetic biomarkers and the underlying molecular mechanism of smoking-related postmenopausal osteoporosis. The difference between the SRPO and control groups was whether osteoporosis occurs or not, which may be related to the smoking‐induced genetic alterations. The genetic biomarkers of SRPO should differentiate SRPO patients and non-osteoporosis postmenopausal female smokers. Thus, we used WGCNA to identify the gene modules that highly correlated to the clinical trait (osteoporosis). In WGCNA algorithm, the number of statistical tests is greatly reduced by clustering the genes into modules and then making statistical analysis. The association between thousands of genes and phenotypes is transformed into that between several gene sets and phenotypes, so as to avoid the problem of multiple hypothesis testing and correction. This processing method strengthens the strong correlation and weakens the weak or negative correlation, making the correlation value more consistent with the characteristics of scale-free network and more biologically meaningful. Taken together, the proper multiple-testing correction has been applied in the WGCNA algorithm. As a result, the blue module (module–trait relationships = 0.88, P-value = 7e-07) was found to have the highest association with osteoporosis. And the blue module was found to have the highest MS value among all of the selected modules. Based on these results, we chose blue module as the research object. 

2.Response: We are very grateful to the reviewer for this helpful suggestion. In order to investigate the biological interaction among the genes in the selected modules from WGCNA, we performed a protein-protein interaction (PPI) network analysis for these genes using the STRING database. After removing the disconnected nodes, there were 998 nodes and 10940 edges in the constructed PPI network. Then, we used the CytoNCA plugin for Cytoscape to perform a topological network analysis in the PPI network. We set the twice median DC value as a threshold to screen significant nodes in the PPI network and generate a subnetwork. Furthermore, nodes where both BC and CC value were greater than the median in the subnetwork were identified as a new core network containing hub genes. Finally, the core network containing 113 hub genes (nodes) and 1831 edges was displayed. The identified hub genes were subjected to enrichment analysis to investigate their biological meaning. 

3.Response: We have made corrections according to the reviewer’s comments. 1) In “Function and pathway enrichment analysis”, a Benjamini–Hochberg adjusted P-value < 0.05 was considered to indicate a significantly enriched GO terms and KEGG pathways. 2) In Fig.5, the “Gene Ratio” was used for statistical significance display. 3) The results of function and pathway enrichment analysis of hub genes indicated that hub genes were mainly concentrated in the regulation of RNA transcription and translation, regulation of cell cycle, ATPase activity, HIF-1 signaling pathway, and NF-kappa B signaling pathway, These GO terms and KEGG pathways seem to have closely association with the development of SRPO. We extended the biological meaning of these terms and their association with SRPO in the discussion. 

4.Response: According to the Reviewer’s suggestion, we extended the reason why we chose SVM-RFE and random forest algorithm for feature genes selection and how these two machine learning algorithms were performed in the “Methods and materials”. In this study, feature genes associated with SRPO were selected using SVM-RFE and RF. SVM-RFE was an efficient feature selection algorithm and had shown promising power in the analysis of the genomics, metabolomics], proteomics, etc. For instance, in the previous study, SVM-RFE had helped identifying biomarkers correlated with breast cancer prognosis (Li J, et.al Tumor Characterization in Breast Cancer Identifies Immune-Relevant Gene Signatures Associated with Prognosis. Front. Genet. 10:1119.). During the performance, the recursive feature elimination approach removed one feature with the smallest weight iteratively to a feature rank until all the features had been removed. In each iteration, the current SVM-RFE model was evaluated by the k-fold cross-validation. Finally, the classifier model with highest accuracy was constructed and the best variables were found. RF had also been widely used for detecting biomarkers in many diseases. The RF algorithm used the variables to construct numerous decision trees and generated the most accurate classes of variables to individual trees. Referring to other similar studies, we used SVM-RFE and RF machine learning algorithms to select feature genes of SRPO in this study. Moreover, we performed a combination strategy to minimize the possibility of losing important features through incorporating genes from two machine learning algorithms. Overlapping the feature genes selected by different machine learning algorithms to identify the most important feature genes is a very common processing method in many previous studies. SVM-RFE and RF machine learning algorithms screened characteristic variables based on different algorithm theories, so the number of feature genes screened out by these two machine learning algorithms were different. We believed that the common feature genes selected by both machine learning algorithms are important genes that are closely related to SRPO. In the following analysis, we found that the six overlapped feature genes show great diagnostic efficiency for SRPO, which indicated the reliability of our research methods. We hope the above explanation can address the reviewer’s comment. 

5.Response: 1) We are very grateful to the reviewer for the helpful comment and we made correction according to the reviewer’s suggestion. In this study, we aimed to investigate potential genetic biomarkers and the underlying molecular mechanism of smoking-related postmenopausal osteoporosis. We collected the gene expression data of 20 postmenopausal female smokers (10 with high BMD and 10 with low BMD). Smoking was the common exposure for these people. The difference between these people was whether osteoporosis occurs or not, which may be related to the smoking‐induced genetic alterations. The genetic biomarkers of SRPO should differentiate SRPO patients and non-osteoporosis postmenopausal female smokers. So, we selected 10 sample of postmenopausal female smokers with high BMD (non-osteoporosis) as the control group, 10 sample of postmenopausal female smokers with low BMD (osteoporosis) as the SRPO group in this study. 2) According to the suggestion of reviewer, we performed multiple-testing correction for the gene expression analysis. In the gene expression analysis of the selected feature genes, the t-test with multiple-testing correction (Benjamini–Hochberg) was used to test for significant differences between the two groups. As a result, the six feature genes, HNRNPC (p=0.0031), PFDN2 (p=0.0013), PSMC5 (p=0.0371), RPS16 (p=0.0202), TCEB2 (p=0.0073), and UBE2V2 (p=4.07e-05) was found statistically significant differentiated between the two groups. 3) According to the reviewer’s suggestion, we rephrased the description and discussion about the functional significance of the implicated genes in the discussion. Through extensive reading of relevant literatures, we explored the biological association between these feature genes and smoking-related postmenopausal osteoporosis, hoping to discover the potential mechanism of these genes in the pathogenesis of SRPO. 

Minor

1.Response: We are very grateful to the reviewer for the comment and we have made corrections to this section. The accurate references were cited in the discussion to link the implicated genes/pathways to SRPO. 

2.Response: We apologize for the language problems in the original manuscript. We have submitted our manuscript to TOPEDIT (https://www.topeditsci.com/) for polishing as suggested by the reviewers. 

3.Response: We are sorry that our English writing level has troubled the reviewers. We have submitted our manuscript to TOPEDIT (https://www.topeditsci.com/) for polishing as suggested by the reviewers. 

4.Response: We are very grateful to the reviewer for this helpful suggestion. We have provided the original gene expression matrix, module information, PPI network, results of GO and KEGG enrichment analysis, and results of machine learning as supplementary files, based on the reviewers' recommendations.

---

## [Decision Letter · Decision Letter 1]

23 Aug 2021

PONE-D-21-08513R1

Analysis of potential genetic biomarkers and molecular mechanism of smoking-related postmenopausal osteoporosis using weighted gene co-expression network analysis and machine learning

PLOS ONE

Dear Dr. Wang,

Thank you for submitting your manuscript to PLOS ONE. After careful consideration, we feel that it has merit but does not fully meet PLOS ONE’s publication criteria as it currently stands. Therefore, we invite you to submit a revised version of the manuscript that addresses the points raised during the review process.

ACADEMIC EDITOR:

The revised manuscript is significantly improved. Reviewer 2's final comments regarding structure/organization should be addressed. The current version of the manuscript presents a highly fragmented narrative, and would benefit from stylistic edits (as suggested by Reviewer 2) to make it more readable.

We look forward to receiving your revised manuscript.

Kind regards,

Jishnu Das, Ph.D.

Academic Editor

PLOS ONE

Journal Requirements:

Additional Editor Comments (if provided):

Reviewers' comments:

Reviewer's Responses to Questions

**Comments to the Author**

1. If the authors have adequately addressed your comments raised in a previous round of review and you feel that this manuscript is now acceptable for publication, you may indicate that here to bypass the “Comments to the Author” section, enter your conflict of interest statement in the “Confidential to Editor” section, and submit your "Accept" recommendation.

Reviewer #1: All comments have been addressed

Reviewer #2: (No Response)

2. Is the manuscript technically sound, and do the data support the conclusions?

Reviewer #1: Yes

Reviewer #2: Yes

3. Has the statistical analysis been performed appropriately and rigorously? 

Reviewer #1: Yes

Reviewer #2: Yes

4. Have the authors made all data underlying the findings in their manuscript fully available?

Reviewer #1: Yes

Reviewer #2: Yes

5. Is the manuscript presented in an intelligible fashion and written in standard English?

Reviewer #1: Yes

Reviewer #2: No

6. Review Comments to the Author

Reviewer #1: The authors have adequately addressed my comments. In particular, I was glad to see that in their Discussion the authors have moderated the significance of their results with the limitations inherent in this initial study.

Reviewer #2: The manuscript has been considerably improved. The authors addressed all technical comments well; the results are now scientifically sound and worth to be published.

One last improvement needed is the writing / organization of the paper. The results section reads too thin to deliver the message fully. The major issue appears to be lacking of relevant discussion/interpretation following specific results. For example, result #2 "Construction of the PPI network" appears insufficient to be an independent section (it's more like a paragraph of Method). An easy fix could be combining it with result #3. Similarly, #5 "Diagnostic efficiency of feature genes" is also too thin - further interpretation is expected, possibly merge with result #4. On the other hand, the authors did a decent amount of work on explaining the specific nominated genes in the Discussion section. I'd suggest move these discussions to the Result section to alleviate the scarcity and better deliver the scientific implication of the analyses.

7. PLOS authors have the option to publish the peer review history of their article (what does this mean?). If published, this will include your full peer review and any attached files.

Reviewer #1: No

Reviewer #2: No

---

## [Author Response · Author response to Decision Letter 1]

26 Aug 2021

Replies to the reviewers’ comments: 

Reviewer #1:

The authors have adequately addressed my comments. In particular, I was glad to see that in their Discussion the authors have moderated the significance of their results with the limitations inherent in this initial study.

Response: We are grateful to Reviewer #1 for his/her effort reviewing our manuscript and his/her positive feedback. The summary of our work as written by this reviewer is precise. Thanks again for your support of our manuscript. 

Reviewer #2:

The manuscript has been considerably improved. The authors addressed all technical comments well; the results are now scientifically sound and worth to be published.

One last improvement needed is the writing / organization of the paper. The results section reads too thin to deliver the message fully. The major issue appears to be lacking of relevant discussion/interpretation following specific results. For example, result #2 "Construction of the PPI network" appears insufficient to be an independent section (it's more like a paragraph of Method). An easy fix could be combining it with result #3. Similarly, #5 "Diagnostic efficiency of feature genes" is also too thin - further interpretation is expected, possibly merge with result #4. On the other hand, the authors did a decent amount of work on explaining the specific nominated genes in the Discussion section. I'd suggest move these discussions to the Result section to alleviate the scarcity and better deliver the scientific implication of the analyses.

Response: We appreciate Reviewer #2 for his/her effort to review our manuscript, and his/her positive feedback. The reviewer gives an accurate summary of our work and brings forward constructive questions. We have addressed them below. According to Reviewers’ suggestion, we combined result #2 with #3. Besides, we agreed with the comments that result #5 "Diagnostic efficiency of feature genes" is thin, and we have moved the explanation of the feature genes in the Discussion to result #5 according to the suggestion. We also extended the relationship of the identified feature genes in our study and smoking-related postmenopausal osteoporosis in the Discussion section. We sincerely hope our responses can address Reviewers’ comments.

---

## [Editor Report · Decision Letter 2]

31 Aug 2021

Analysis of potential genetic biomarkers and molecular mechanism of smoking-related postmenopausal osteoporosis using weighted gene co-expression network analysis and machine learning

PONE-D-21-08513R2

Dear Dr. Wang,

We’re pleased to inform you that your manuscript has been judged scientifically suitable for publication and will be formally accepted for publication once it meets all outstanding technical requirements.

Kind regards,

Jishnu Das, Ph.D.

Academic Editor

PLOS ONE
---

## [Editor Report · Acceptance letter]

15 Sep 2021

PONE-D-21-08513R2 

Analysis of potential genetic biomarkers and molecular mechanism of smoking-related postmenopausal osteoporosis using weighted gene co-expression network analysis and machine learning 

Dear Dr. Wang:

I'm pleased to inform you that your manuscript has been deemed suitable for publication in PLOS ONE. Congratulations! Your manuscript is now with our production department. 

Kind regards, 

on behalf of

Dr. Jishnu Das 

Academic Editor

PLOS ONE